# MOTION PLANNING TRANSFORMERS: ONE MODEL TO PLAN THEM ALL

## ABSTRACT

Transformers have become the powerhouse of natural language processing and recently found use in computer vision tasks. Their effective use of attention can be used in other contexts as well, and in this paper, we propose a transformer-based approach for efficiently solving complex motion planning problems. Traditional neural network-based motion planning uses convolutional networks to encode the planning space, but these methods are limited to fixed map sizes, which is often not realistic in the real-world. Our approach first identifies regions on the map using transformers to provide attention to map areas likely to include the best path, and then applies traditional planners to generate the final collision-free path. We validate our method on a variety of randomly generated environments with different map sizes, demonstrating reduction in planning complexity and achieving comparable accuracy to traditional planners.

## 1 INTRODUCTION

Motion planning is an integral component of any 2D navigation stack. It is an age-old problem with more than a few decades of innovative solutions. By far the most popular recent methods involve constructing trees or graphs in the planning space by randomly sampling points (LaValle & James J. Kuffner, 2001; Kavraki et al., 1996) or discretizing the space into grids (Hart et al., 1968).

Although these methods provide theoretical guarantees, they do not scale well to problems with larger planning spaces and crowded environments. This is a frequent problem for robots in large warehouses and household cleaning robots. Recently, learning-based methods that leverage environment structure from the given data have been proposed for these kind of problems. They combine convolution neural networks (CNN)

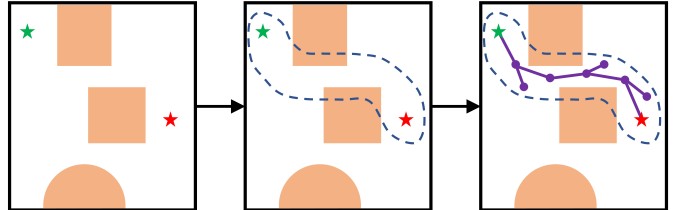

Figure 1: Given a planning problem (left), we propose a hierarchical approach to planning. In our method, the MPT proposes a region where a valid path might exist (center). A sampling-based planner is used to find a collision-free path (right) given the proposed region.

and multi-layer perceptrons (MLP) to predict where samples should be generated to construct trees (Qureshi et al., 2019; Johnson et al., 2020; Kumar et al., 2019), but do not extend to maps of larger sizes. Methods such as Value Iteration Network (VIN)(Tamar et al., 2016), which do extend to maps of different sizes, perform poorly with larger maps (Nardelli et al., 2018). Hence, in our ever-expanding pursuit of automation, we need planning algorithms that can scale up while effectively keeping the planning problem's complexity and computation time in check.

The trajectory of a local plan is influenced by the orientation of far-away obstacles, similar to language models where the semantics of a sentence is inferred only by reading the entire sentence. The ability of transformers to understand the syntactic and semantic structure of sentences was shown in Vaswani et al. (2017). Thus the capacity of transformers to learn such connections efficiently is the motivation of this work. Our model plans in a two-stage approach (Fig. 1), first a transformer-based region proposal network recommends a region where a potential path can exist. We refer to this module as the Motion Planning Transformer (MPT). Next, a sampling-based motion planner (SMP) tries to plan

a path given the proposed region. Since the inputs to the transformers are sets rather than vectors, maps of different sizes can be used, and by reducing the search space, SMP can generate solutions much faster with fewer nodes.

We assess the performance of MPT augmented SMPs (Karaman & Frazzoli, 2011; Gammell et al., 2014) on a synthetic and real-world dataset comprising different map sizes in cluttered environments for point-mass and non-holonomic car models. Our results indicate that MPT augmented planners achieve 7-28% improvement in accuracy over recent learning-based planners, while matching the accuracy to traditional planners. MPT planners also reduce both planning time by 7-25 times and the vertices on the planning tree by 2-12 times compared to traditional planners. We also propose a new training routine to allow for MPT to generalize to maps of different sizes while maintaining accuracy and reducing both planning time and vertices in the planning tree. We also show that MPT maintains the same qualities to real-world environments without fine-tuning or additional training. The diversity of environments represented, from block obstacles to narrow passage mazes speak to the potential general application of the MPT approach across multiple environment types and domains.

## 2 RELATED WORK

The most relevant work to our transformers-based region proposal network for motion planning is perhaps the guided sampling-based motion planning methods. They analytically or through learned heuristics determine a subset in robot space that probably contains a path solution. For instance, (Qureshi & Ayaz, 2016; Tahir et al., 2018) employ Artificial Potential Fields (APF) within sampling-based methods such as RRT* (Karaman & Frazzoli, 2011) and Bidirectional RRT* (Qureshi & Ayaz, 2015) to guide a subset of random samples towards promising regions that possibly contain an optimal path solution. In contrast, Informed-RRT* (IRRT*) (Gammell et al., 2014) depends on an initial path from an RRT* algorithm to compute an ellipsoidal region probably containing an optimal path solution. However, in most planning problems finding an initial path solution is itself challenging. In a similar vein, Batch Informed Trees (BIT*) (Gammell et al., 2015) begins from an elliptical region formed by a straight line path ignoring all obstacles and incrementally expand it until an initial path solution is found. Once an initial path is determined, it is further optimized by adapting the precomputed ellipsoid and generating new samples within that space. Despite all advancements, these methods only consider geometric planning, i.e., path planning under collision-avoidance constraints. They are yet to be evaluated in practical problems with complex constraints such as kinodynamic or non-holonomic constraints for autonomous car navigation tasks.

Many have also used learning-based methods to reduce search spaces. Williams (1992) used the REINFORCE algorithm to guide the underlying SMP planner on a discretized workspace, others use a learned value function to guide the graph search (Choudhury et al., 2018; Chen et al., 2020), while Kumar et al. (2019) learned a generative model to sample points along bottleneck regions. Similarly, Value Iteration Networks (VIN) (Tamar et al., 2016) also discretizes the space and learns a value map to guide path planning. Universal Planning Networks (UPN) (Srinivas et al., 2018) extends VIN to continuous control spaces. Chaplot et al. (2021) proposed one of the few works that use transformers to learn a value function. Other works employ latent representation of state-spaces to generate a plan (Banino et al., 2018; Emmons et al., 2020; Ichter & Pavone, 2019). These methods are often difficult to train and interpret, and many of them are yet to be evaluated in real world navigation tasks.

Although there exist random sampling-based approaches such as RRT* (Karaman & Frazzoli, 2011; Arslan et al., 2017) and SST (Li et al., 2016) that explore the robot state space and satisfy advance constraints, they suffer from the curse of dimensionality and take a considerable computational time in cluttered spaces. Neural Motion Planning (Qureshi et al., 2020; Ichter et al., 2018) has recently emerged as a promising tool for solving a wide range of planning problems under various task constraints, ranging from non-holonomic (Johnson et al., 2020; Li et al., 2021) to advanced manifold kinematic constraints (Qureshi et al., 2020), with high computational speed. These methods learn sampling distributions from expert demonstrations and, on execution, generate samples for an underlying planner forming a subset that potentially contains a path solution. However, these approaches assume a fixed size input environment map and often require redefining network architectures and retraining for different map sizes. However, recent developments in deep learning, primarily through Transformers (Dosovitskiy et al., 2021; Liu et al., 2021), have provided us with ways to relax such assumptions. Our proposed approach leverages these developments and introduces a region proposal

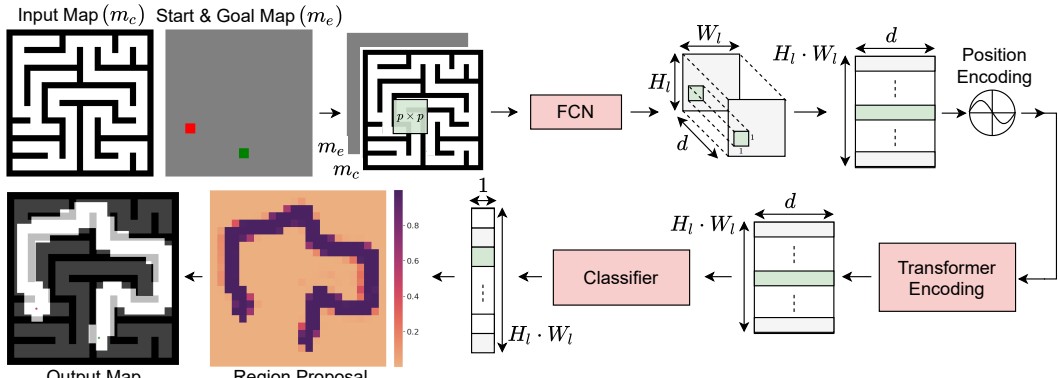

Figure 2: Overview of MPT Module. (Start from the top left and move clockwise) The input map, and the start (green) and goal (red) encoded map are concatenated together and passed as inputs to the model. The Fully Convolution Network (FCN) passes a sliding window of size $p \times p$ over the input and encodes each patch into latent vectors of dimension $d$. After reshaping, fixed positional encodings are added to the latent vectors to inject spatial location. The modified latent representation is used by the transformer module to identify patches through which a path might exist. This information is encoded into a latent vector of size $d$, which is used by a classifier to provide the probability that a path might pass through the patch. The patches with probability greater than 0.5 are used to create the mask for the map. The light green shading highlights the flow of information for a single patch from the input to the output of the model.

framework that can work with variable map sizes and enhances underlying motion planners to solve complex problems in cluttered environments.

## 3 METHOD

The set of motion planning problems that MPT aims to solve is of 2D navigation. Given a start ($x_s$) and goal ($x_g$) state, the objective is to propose a sequence of states $x_i \in \mathcal{X}$ for $i \in \{0, 1, \ldots n\}$ such that $x_0 = x_s$, $x_n \in \mathcal{X}_{\text{goal}}$ and the trajectories joining these states should be both kinematically feasible and collision free. Here, $\mathcal{X}$ represents the robot state space and $\mathcal{X}_g \triangleq \{x : \|x - x_g\| \leq \epsilon, x \in \mathcal{X}\}$ for a user defined threshold ($\epsilon$). Since we focus on solving the motion planning problem for 2D mobile systems, $\mathcal{X} \in \{\mathbb{R}^2, SE(2)\}$. MPT proposes promising regions in $\mathcal{X}$ where the underlying planners can search for path solutions. In the following sections we describe our method in detail.

### 3.1 MOTION PLANNING TRANSFORMER (MPT)

The MPT module is a region proposal network, similar to Faster R-CNN (Ren et al., 2015), but uses a transformer network to identify regions of interest. An overview of the model is shown in Figure 2. The input to the model is a representation of the planning scene, and an encoding of the start and goal points. The planning scene is represented using an occupancy matrix, $\boldsymbol{m_c} \in \mathbb{R}^{H \times W}$, where an element with 1 indicates an occupied space and 0 denotes free space. In some navigation problems, these representations are also called costmap and have additional cost terms associated safety and robot constraints. The start-goal encoding of the planning problem is formed by highlighting patches of size $p \times p$ on a tensor of size $H \times W$ with values -1 and 1 for the start and goal point respectively. These two matrices are concatenated to form a tensor $\boldsymbol{m}$ and passed to the feature extractor.

**Feature Extractor:** The feature extractor is a Fully Convolution Network (FCN) that encodes the environment and the given planning problem into a latent space. As shown in prior works (Dosovitskiy et al., 2021; Ren et al., 2015), the feature extractor reduces the dimensionality of the input space by using a series of convolution, ReLU, and MaxPool layers. The FCN passes a sliding window of size $p \times p$ over $\boldsymbol{m}$ to generate an output of size $H_l \times W_l \times d$, where $H_l$ and $W_l$ is determined by the size of the costmap and the FCN, and $d$ is the latent dimension of the transformer encoder. Each patch that the sliding window moves over is parameterized by an anchor point similar to Faster R-CNN (Ren et al., 2015). We choose the 2D co-ordinate corresponding to the center pixel of the patch as

the anchor point. The output of the FCN is reshaped to size $(H_l \cdot W_l) \times d$ and fed to the position encoder. Each row vector of this matrix corresponds to an anchor point on the input map.

**Position Encoding:** Transformer and convolutional models are agnostic to the spatial location of their inputs Dosovitskiy et al. (2021). A common solution is to add learned or fixed vectors to encode the position of each input (Gehring et al., 2017). For testing our model we used the same position encoding as Vaswani et al. (2017), but for training, we use the following position encoder:

$$PE(j, 2i) = \sin\left(\frac{k+j}{10000^{2i/d}}\right) \qquad PE(j, 2i+1) = \cos\left(\frac{k+j}{1000^{(2i+1)/d}}\right)$$

where $k$ is uniformly sampled from a discrete set of integer numbers $\mathcal{Z}$, $j$ is the index of the anchor point, and $i$ is the index of latent vector. This modification enables us to generalize to maps of different sizes. For maps larger than the training data, we observed that the position encoder from Vaswani et al. (2017) created a bias that prevented MPT from selecting regions of the state space outside the training region. To overcome this bias, we leverage the fact that a proposed plan is invariant to linear translation of the state space, and train our model by randomly shifting the position encoder. This ensures that the model is not biased by the position encoder. For further details refer to Section C in the Appendix. The modified latent vector is then passed to the transformer encoder.

**Transformer Encoder:** The transformer module is responsible for learning the connections between the different local regions on the map for a given planning problem. It infers these connections by passing the latent vectors through a series of multi-headed self-attention (MSA) and multi-layer perceptron (MLP) blocks. Between every MSA and MLP block we apply Dropout, Layer Norm and residual connections similar to Vaswani et al. (2017) and Dosovitskiy et al. (2021). We also add gradient checkpoints (Chen et al., 2016) after the MSA blocks to be more memory efficient. Hence by attending to all patches, the network encodes the importance of each patch to the given planning problem in its output.

**Classifier:** Using the encoded importance, the classifier predicts if each of the anchor point is of interest to the current planning problem. This is efficiently implemented using a $1 \times 1$ convolution layer, similar to Faster R-CNN (Ren et al., 2015). A mask is generated by setting $p \times p$ patch, centered around selected anchor points, to 1 on a matrix of size $H \times W$.

## 3.2 PATH PLANNING

Any traditional or learning-based planner can be used to find the path by searching the masked region. In this work, we use variations of the Rapidly Exploring Random Trees (RRT) algorithm, that guarantee optimality (Karaman & Frazzoli, 2011; Li et al., 2016; Gammell et al., 2014), to find the path.

**Hybrid planning:** In rare occasions, due to mis-classifications, MPT guided planners fail to generate a solution. To overcome these errors, we alternate between searching the masked (exploitation) and unmasked (exploration) regions of the map. We show that this technique is able to preserve the benefits of MPT guided planners while achieving higher efficiency in planning.

## 4 EXPERIMENTS AND RESULTS

We evaluated the planning capabilities of MPT aided SMPs and compared them with both traditional and learning-based planners. In the following section, we will go over our environment and robot setup, model training, and results.

## 4.1 SETUP

**Environments:** To test the planning capabilities of our method, we evaluated the model on randomly generated maps from two different classes of environments. The first environment is called the Random Forest, where circular and square objects are randomly placed on the map. It simulates real world scenes with narrow passages and crowded spaces. The second environment is called the Maze environment. Each map is a perfect maze, generated using randomized depth-first search. A characteristic of a perfect maze is that any start and goal pairs on this map are reachable by a

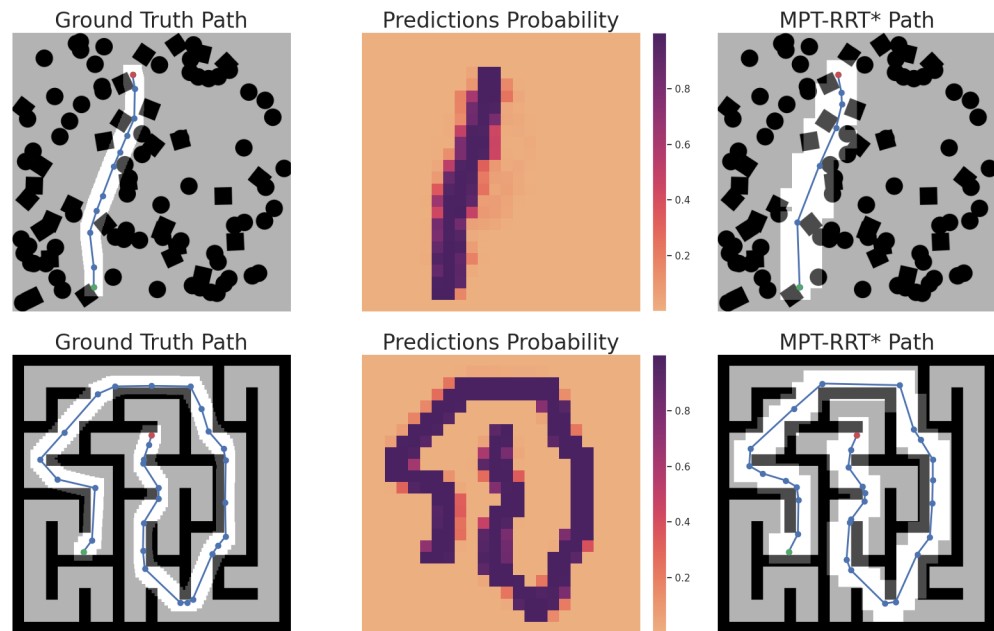

Figure 3: Planned paths for two different planning problems. Top Row: Random Forest environment. Bottom Row: Maze environment. Left Column: The ground truth path for a given start (green) and goal (red) point from the validation data. Center Column: The prediction probability from the MPT. Right Column: The masked map and the planned trajectory for the given start (green) and goal (red) point using RRT*.

collision-free path. These maps mimic long-horizon planning problems because even if the start and goal are geometrically close, the planner would have to take a circuitous path to reach the goal. We also tested our method on the map of building 079 University of Freiburg constructed from publicly available 2D lidar scan data.

An integral component of any motion planner is the ability to check if the robot is in collision with any obstacles. This can be accomplished using geometric methods such as the Gilbert–Johnson–Keerthi (GJK) algorithm or other learning based approaches (Chase Kew et al., 2021). We use the occupancy map to check if the robot is in collision with the obstacle; hence we use a high resolution of 5cm per pixel for our maps.

**Robot Models** We test our algorithm on two kinds of robotic systems that encapsulate a large set of mobile systems. The first is a simple Point Robot Model that can move in any direction in its state-space $\mathcal{X} = \mathbb{R}^2$. Solutions for this model can be easily extended to a large number of indoor and outdoor robots. A simple example is robots with circular footprints. For such robots, inflating the obstacles by the robot's radius reduces its footprint to a point (Choset et al., 2005). It is also common in 2D navigation to bound the non-circular footprints using a circle for path planning (van den Berg et al., 2011; Marder-Eppstein et al., 2010). The other robot we tested is a Dubins Car Model with state-space $\mathcal{X} = SE(2)$, controlled using wheel velocity and steering angle. This robot model represents a class of mobiles systems such as cars, bikes and even needle-steering in surgeries (Alterovitz et al., 2008).

## 4.2 TRAINING

The MPT model is trained in an end-to-end fashion under supervision similar to Girshick (2015). Each mini-batch is formed from a single planning problem containing positive and negative anchor points. We define positive anchor points as those within 0.7m distance to the trajectory, while all the others are considered negative samples. For training, negative anchor points are randomly chosen such that we at least have a ratio of 1:1 of positive and negative samples. For the Dubins Car Model, we considered only the robot's 2D coordinates to identify the positive and negative anchor points.

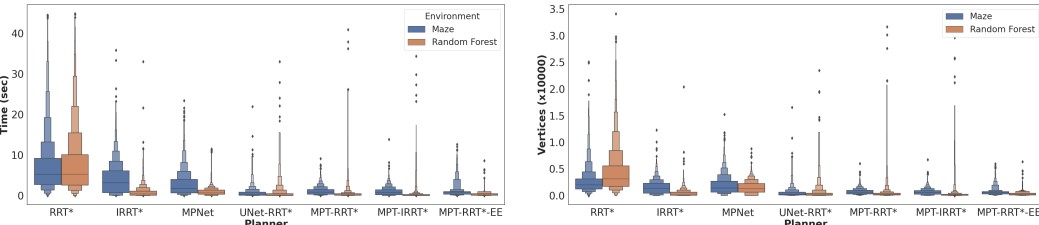

Figure 4: (From left) Path planned by RRT*, IRRT*, MPT-RRT*, and MPT-IRRT* for the same start (green) and goal (red) positions. MPT aided planners are able to reduce significantly the number of vertices (orange) required to search for a path.

Figure 5: Planning statistics for the Point Robot Model. Left: The planning time for traditional and learning-based planners. Right: Number of vertices in the planning tree for traditional and learning-based planners. MPT aided planners consistently reduce the planning time and the vertices in the planning tree, resulting in a lower variance of these statistics for these planners.

We trained the network by minimizing the cross-entropy loss using the Adam optimizer (Kingma & Ba, 2015) with $\beta_1 = 0.9$, $\beta_2 = 0.98$, and $\epsilon = 1e^{-9}$. We varied the learning rate as proposed in Vaswani et al. (2017) with warm-up steps of 3200. Each model was trained for 1000k epochs with a batch size of 128. The models were trained on one machine with 4 NVIDIA 2080GTX graphics card. The MPT model for the Point Robot took 21hrs and for the Dubins Car took 12hrs to train.

**Datasets:** We trained an MPT model for the Point Robot on 1750 randomly generated maps from each environment mentioned above. For each map, 25 paths were generated using the RRT* planner. Similarly we trained an MPT model for the Dubins Car on 1000 randomly generated maps from the Random Forest environment. We collected 50 paths from each map using the SST planner (Li et al., 2016). All the maps used were of size $480 \times 480$.

### 4.3 POINT ROBOT MODEL

For the Point Robot Model, we compared MPT-aided planners with traditional and learning-based planners. We also looked at the capabilities of other image segmentation approaches such as UNet (Ronneberger et al., 2015) to highlight the area where a potential path might exist. We choose UNet because, like MPT, it can generalize to maps of different sizes. We call the aided planners Y-X, where Y is the underlying method used to generate the mask and X is the SMP planner. MPT-RRT*-EE represents the MPT aided RRT* planner with the exploration and exploitation strategy (hybrid planning).

The first set of experiments examined the network's ability to generalize to unseen maps of the same dimension as the training data. We compared the planners on 2500 random start and goal pairs for maps from the Maze and Random Forest environment. For each planner, we report the 1. Accuracy - the percentage of planning problems the planner solves, 2. Time (sec) - the amount of time it takes to generate the mask (if applicable) and plan a path shorter than a path from the RRT* planner searching for a fixed time, and 3. Vertices - the number of collision-free states sampled by the planner to construct the planning tree. The summary statistics of the experiment are reported in Table 1. We see that MPT aided planners reduce the planning time and vertex count of the planning tree substantially. In Fig. 3, we show two examples of a planned path from this experiment. In Fig. 4 we show the vertices used by the planning tree for various planners for a single planning problem.

Table 1: Comparing planning accuracy, and median time and vertices for the Point Robot Model on unseen environments of the same size as the training data.

| Algorithm | Random Forest | | | Maze | | |
|---|---|---|---|---|---|---|
| | Accuracy | Time (sec) | Vertices | Accuracy | Time (sec) | Vertices |
| RRT* | 100% | 5.448 | 3228 | 100% | 5.364 | 2042 |
| IRRT* | 100% | 0.425 | 267 | 100% | 3.139 | 1394 |
| UNet-RRT* | 30.27% | 0.167 | 168 | 21.4% | 0.346 | 277 |
| UNet-RRT*-EE | 100% | 2.58 | 1913 | 100% | 4.133 | 2139 |
| MPNet | 92.35% | 0.296 | 63 | 71.76% | 1.727 | 1409 |
| MPT-RRT* (ours) | 99.40% | 0.194 | 233 | 99.16% | 0.870 | 626 |
| MPT-IRRT* (ours) | 99.40% | 0.087 | 136 | 99.16% | 0.784 | 566 |
| MPT-RRT*-EE (ours) | 100% | 0.211 | 247 | 100% | 0.869 | 585 |

To better understand the advantages of MPT, we visualize the distribution of the planning time and vertices in Fig. 5 for the Random Forest and Maze environment using Letter-value plots (Hofmann et al., 2011). These plots help to observe the tail of the distribution of the metrics. Naive RRT* has a heavier tail distribution than MPT-RRT* because for start and goal pairs further away from each other, the planner needs to generate a denser graph to search a larger space, requiring more time and vertices. On the other hand, MPT-aided planners focus their search near regions highlighted by the model, and as a result, they plan faster with fewer vertices. We see a thin tail distribution for MPT-RRT* and MPT-IRRT* planners for the random forest environment. This is because the planner only terminates when the cost of the path is below a certain threshold. Even if a solution is found, MPT planners continue to search to reduce the path length, resulting in longer planning times and vertex count for few trajectories. The number of such problems accounts for less than 0.75% of the planning problems.

We also observe that IRRT* performs considerably better than RRT* in the Random Forest environment and achieves similar planning time and planning tree vertices compared to the aided planners. This is because IRRT*, like MPT's, reduces the planning search space once an initial solution is found by bounding the initial path with an Ellipse. Such heuristics do not work for long-horizon problems like the Maze environment, and MPT aided planners outperform traditional methods.

The MPT model also outperforms other learning-based approaches. Planners that used UNet to propose patches performed poorly. We believe this is because the convolution layers can only learn the connections between local patches, and deeper networks would be required to learn global connections. As a result, it fails to highlight the area of interest for the given planning problem. MPNet, on the other hand, performs

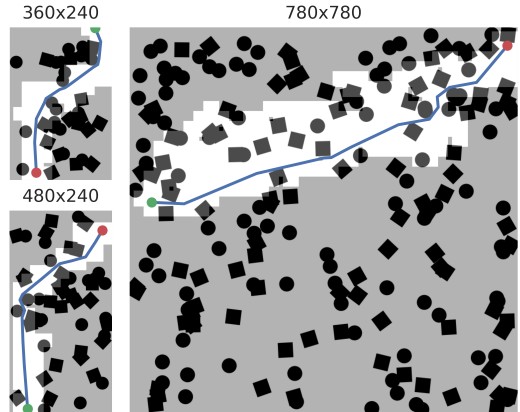

Figure 6: Plot of paths for Random Forest environments of different size. The architecture of the MPT Model allows flexibility in planning for environments of different sizes.

considerably better than UNet-RRT* for both environments, but not as well as MPT planners. We attribute the weak generalization to the lack of training data. In Qureshi et al. (2019), the authors used nearly 400k trajectories to train their model, whereas we only provided around 88k trajectories for all our models.

Due to classification errors, MPT fails on around 1% of maps (Some of these paths are plotted in the Appendix). Nevertheless, by randomly exploring the map for few samples outside the segmented region, MPT-RRT*-EE can solve these problems while UNet-RRT*-EE has similar statistic to RRT*. The distribution of planning time and vertices for MPT-RRT*-EE are much tighter compared to traditional planners and even MPT planners without exploration.

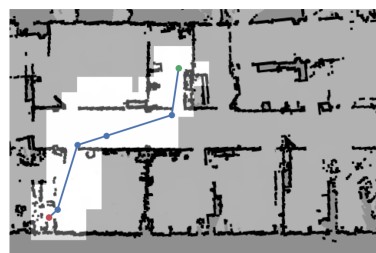
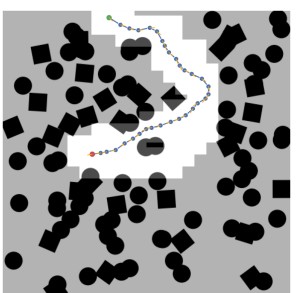
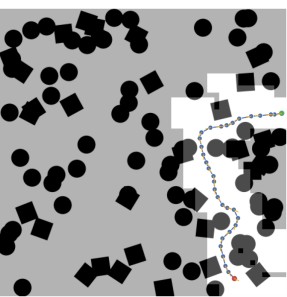

Figure 7: Path generated by MPT-RRT* on a random start and goal pair on the map of building 079 at University of Freiburg using a model trained only on simulated environments.

Figure 8: MPT can also be trained to aid SMP planners for non-holonomic robots. Left and Right: Planned paths on Random Forest environment using MPT-SST. MPT identifies regions through which a non-holonomic path exists.

The following experiment we conducted evaluated MPT's ability to generalize to map sizes that were not part of the training data. We tested the model on four different map sizes, $360\times240$, $480\times240$, $560\times560$, and $780\times780$ of the Random Forest environment on 1000 randomly generated maps while maintaining the density of obstacles. We used the same MPT model without any re-training or fine-tuning and compared the same metrics with traditional planning techniques. We did not compare against learning-based planners because these planners either performed poorly in our previous experiment or were not generalizable to maps of different sizes without modifications. The planning statistics are summarized in Table 2, and three successfully planned paths using MPT are shown in Fig. 6. Training the model by randomly shifting position encoding was instrumental in improving the accuracy of the planner for maps of larger sizes. Without it, the planner was able to achieve only 75.6% accuracy on maps of size $560 \times 560$. It also improved overall accuracy by 1-4% on other maps as well. The complete statistics are reported in the Appendix C.

We notice that the MPT aided planners, similar to our previous experiments, achieve lower planning time and vertices count than traditional planners without additional training or fine-tuning. IRRT* achieves similar performance to MPT-aided planners for smaller-sized maps. As the map sizes grow, the time taken by IRRT* grows because of the more prominent search space. While for larger maps, MPT aided planners can find a solution faster and outperform IRRT*.

The final experiment we performed was to test the MPT model on a real-world map. We obtained a map of building 079 University of Freiburg from publicly available 2D lidar scan data. We again used the MPT model from our previous experiments without any additional training or fine-tuning. The results are summarized in Table 3. We plot one of these trajectories in Fig. 7. The ability of MPT to generalize to real world maps without any additional training or fine tuning is exciting because it can be trained on simulated environments and transferred to real world applications with minimal effort.

### 4.4 DUBINS CAR MODEL

To examine if the MPT can aid in the planning of non-holonomic systems, we trained a new MPT model to plan for a Dubins Car robot. We tested the trained model on 1000 Random Forest environments and compared the metrics with those from the naive SST planner. The metrics we report is similar to the point robot, but the time we report is the interval that the planner takes to find a valid path. SST aided by MPT is called MPT-SST and the hybrid planner is called MPT-SST-EE. Two trajectories from this experiment is plotted in Fig. 8. We can observe that the MPT highlights regions so that SST can generate a kinematically feasible path. Planners such as SST, which sample in the control space, do not scale well to larger planning spaces, and MPT can help with reducing the search space. The timing results reported in Table 4 support this claim. MPT helps in reducing the planning time and vertices for the underlying SST planner.

## 5 DISCUSSION

Graph-based search methods such as A* can also be used to solve the given planning problem using the costmap, but in order to do real time planning, these methods often require sub-sampling of the

Table 2: Comparing planning accuracy, and median time and vertices for Point Robot on maps of the different sizes of the Random Forest environment.

| Map Size (# Obstacles) | | RRT* | IRRT* | MPT-RRT* | MPT-IRRT* | MPT-RRT*-EE |
|---|---|---|---|---|---|---|
| 360×240 (35) | Accuracy | 100% | 100% | 99.20% | 99.20% | 100% |
| | Time (sec) | 5.926 | 0.286 | 0.248 | 0.054 | 0.297 |
| | Vertices | 3660 | 257 | 354 | 106 | 382 |
| 480×240 (50) | Accuracy | 100% | 100% | 98.5% | 98.5% | 100% |
| | Time (sec) | 6.308 | 0.394 | 0.265 | 0.073 | 0.302 |
| | Vertices | 3480 | 291 | 319 | 131 | 362 |
| 560×560 (100) | Accuracy | 100% | 100% | 99.7% | 99.7% | 100% |
| | Time (sec) | 6.725 | 0.283 | 0.181 | 0.083 | 0.217 |
| | Vertices | 3854 | 203 | 218 | 112 | 237 |
| 780×780 (200) | Accuracy | 100% | 100% | 99.5% | 99.5% | 100% |
| | Time (sec) | 8.095 | 0.476 | 0.297 | 0.152 | 0.285 |
| | Vertices | 4292 | 255 | 241 | 161 | 238 |

Table 3: Comparing planning accuracy, and median time and vertices for the Point Robot Model on real world map.

| Planner | Accuracy | Time (sec) | Vertices |
|---|---|---|---|
| RRT* | 20/20 | 2.507 | 1630 |
| MPT-RRT* | 17/20 | 0.538 | 455 |
| MPT-RRT*-EE | 20/20 | 0.946 | 804 |

Table 4: Comparing planning accuracy, and median time and vertices for Dubins Car Model for the Random Forest Environment.

| Planner | Accuracy | Time (sec) | Vertices |
|---|---|---|---|
| SST | 100% | 3.313 | 1200 |
| MPT-SST | 87.6% | 2.617 | 932 |
| MPT-SST-EE | 100% | 2.712 | 1150 |

costmap to provide solutions in real time. Further hand-tuned heuristics are required to restrict the planning space for kinematically constrained systems (Dolgov et al., 2010) otherwise they get stuck in local minima (LaValle & James J. Kuffner, 2001). We have shown through our experiments that the MPT aided planners require no such sub-sampling, and can learn to restrict search spaces by leveraging data. Since MPT is agnostic to the underlying planner, graph-based searches can also be used to search the highlighted space.

Prior learning based methods that predicts a value such as VIN (Tamar et al., 2016), are shown to be difficult to train for larger maps (Nardelli et al., 2018) because of the depth of layers that is required to propagate the reward. Unlike the other transformer based planner (Chaplot et al., 2021), as we have shown randomizing the positional encoding is critical to be able to generalize the planner to maps of different sizes. The map sizes that are used in our method is nearly 10 times larger than the ones shown use in Chaplot et al. (2021). Moreover, MPT combines best of both worlds by learning from data the space where a plan exists and using an optimal planner get the exact path.

## 6 CONCLUSION

In the future, extending MPT to higher dimensional robots such as robotic arms or drones is an interesting problem with many real-world applications. For mobile robots in 3D space, the use of sparse Transformer models(Roy et al., 2021; Beltagy et al., 2020) would be better suited because of the dimensionality of the space. While for manipulation systems in 3D since the task and joint space do not overlap, the extension of method is more challenging.

In this work, we have shown an application of the transformer model for 2D navigation tasks. Unlike prior methods that need to retrain models for maps of different sizes, we leverage the ability of transformers to handle sequences of different lengths and parallelize long-term dependencies without recursions for planning problems with different map sizes. By combing MPT with SMP, we could generate paths faster, with fewer tree nodes for different environments and robot models.

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

## A  APPENDIX

## B  NETWORK ARCHITECTURE

### B.1  MPT

In this section, we detail the network architecture used in for MPT in our experiments. The Transformer architecture is similar to the ones proposed in Vaswani et al. (2017). We used 6 layers of encoder block, each consisting of 3 heads. The dimension of the keys and queries was set at 512, and the dimension of the value was set at 256. The architecture of our feature extractor is given in Table 5. For the convolution layer, the dimensions in the brackets represent [Input Channel Size, Output Channel Size, Kernel Size, Stride], and for the Maxpool layer, it represents the Kernel Size

Table 5: Network architecture of the Feature Extractor

| Layer | Dimension |
|---|---|
| 2D Convolution | [2, 6, 5, 0] |
| 2D Maxpool | [2] |
| ReLU | |
| 2D Convolution | [6, 16, 5, 0] |
| 2D Maxpool | [2] |
| ReLU | |
| Convolution | [16, 512, 5, 5] |

### B.2  UNET

The UNet architecture we use is similar to the one used by Ronneberger et al. (2015). The network consists of 4 blocks of down convolution, and 4 blocks of up convolution. Each down convolution block consists of a $3 \times 3$ convolution, followed by batch norm, ReLU and a $2 \times 2$ of max pool layer. The up convolution block consists of a bilinear up sampling block, followed by a $3 \times 3$ convolutions, followed by batch norm and ReLU. The final layer is a $1 \times 1$ to classify each latent vector. For more details for the exact channel size used for each convolution please refer to our code.

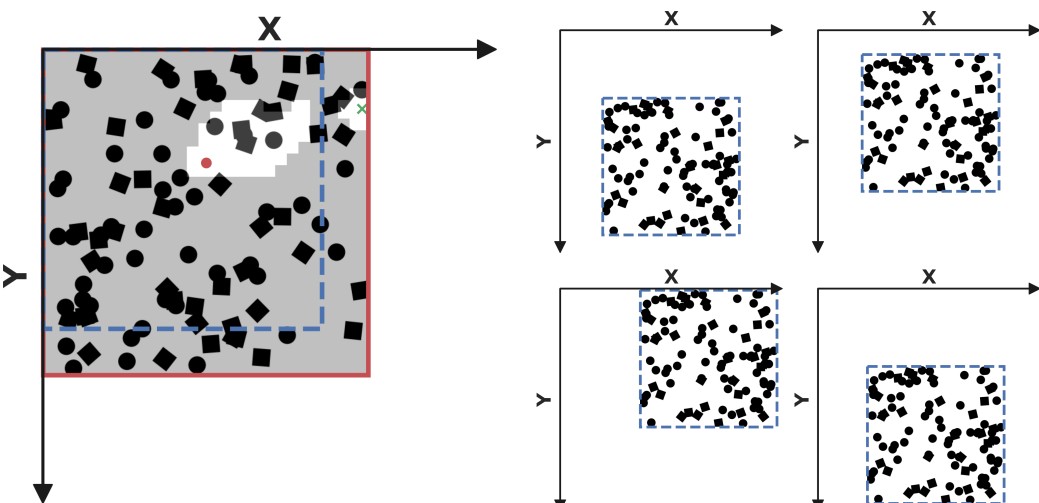

Figure 9: To encode the $(x, y)$ position of each pixel in the world co-ordinate we add fixed position vectors. Left: Region proposed by an MPT model trained with position encoding proposed in Dosovitskiy et al. (2021) for a map larger than the training dataset. The blue box marks the size of the training dataset and the red box marks the size of test dataset. The model fails to predict regions near the boundary of the training map. Right: A training map, randomly shifted in world coordinates. Randomly adding offsets to position encoding has the same effect as translating the map in world co-ordinates.

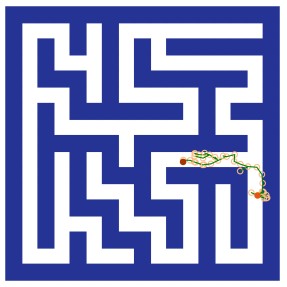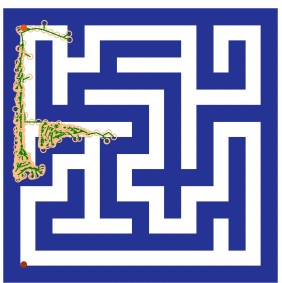

Figure 10: Plots of two paths for the modified maze environment using the NEXT-KS planner. The yellow circles indicate the sampled points by the planner. The planner is able to solve for simple problems (Left) while for long horizon problems it gets stuck in local minimums.(Right)

Table 7: Comparing planning accuracy, planning time, and number of vertices in the tree for the Point Robot on a down sampled Maze Environment

| Algorithm | NEXT-KS |
|-----------|---------|
| Accuracy | 28.28% |
| Time (sec) | 3.021 |
| Vertices | 387 |

## C  POSITION ENCODING

We observed that the MPT model trained on the position encoding proposed by Vaswani et al. (2017) showed signs of overfitting. The model fails to highlight the region near the boundary of the training map size (See Fig.9 (Left)). One way to resolve this overfitting is to train on maps of various sizes to ensure that the network observes different position encoding. However, this procedure is often computationally expensive, especially for larger maps. We propose to randomly translate the planning space by randomly offsetting the position encoding values. The MPT model trained with the randomized position encoding overcomes the overfitting. Results for the model trained without randomized position encoding are given in Table 2. We observe that the MPT model trained with randomized position encoding perform better overall, but for larger maps achieve nearly 24% more accuracy.

Table 6: Comparing planning accuracy, planning time, and number of vertices in the tree for Point Robot on maps of the different sizes of the Random Forest environment for model trained without random shifting of positional encoding.

| Map Size (# Obstacles) | | MPT-RRT* | MPT-IRRT* |
|------------------------|-----------|----------|-----------|
| 360×240 | Accuracy | 97.4% | 97.4% |
| | Time (sec) | 0.265 | 0.062 |
| (35) | Vertices | 377 | 118 |
| 480×240 | Accuracy | 96.3% | 96.3% |
| | Time (sec) | 0.268 | 0.072 |
| (50) | Vertices | 348 | 129.5 |
| 480×480 | Accuracy | 98.96% | 98.96% |
| | Time (sec) | 0.235 | 0.092 |
| (100) | Vertices | 251 | 133 |
| 560×560 | Accuracy | 75.6% | 75.6% |
| | Time (sec) | 0.253 | 0.082 |
| (100) | Vertices | 262 | 101 |

## D  COMPARISON WITH NEXT PLANNER

To provide a context for value-based planners in solving long horizon problems we compared it against the recently proposed Neural Exploration-Exploitation Tree with kernel smoothing (NEXT-KS) (Chen et al., 2020) on a simplified Maze environment. The same validation set from Table 1 was used, but the map size was reduced from 480×480 pixels to 16×16 pixels. The results for this experiment are summarized in Table 7. The drop in performance can be attributed to the value function proposing samples that are locally optimum. See Fig 10 (Right) for an example of a failed plan. In addition, because of the recursive nature of the planner, the timing for planning is similar to that of IRRT*. We do not compare the planners performance on the forest environment because reducing the map size reduces the distance resolution from 0.05meter/pixel to 0.75meter/pixel, which if applied to the Forest environment would oversimplify collision free regions.

# E    SUCCESSFUL TRAJECTORIES

In this section we plot few more successful trajectories of fixed map sizes for the Random Forest and Maze environments in Fig. 12 and 14 respectively. In Fig. 11 and 13 we plot the vertices sampled by the different planners to construct their planning trees.

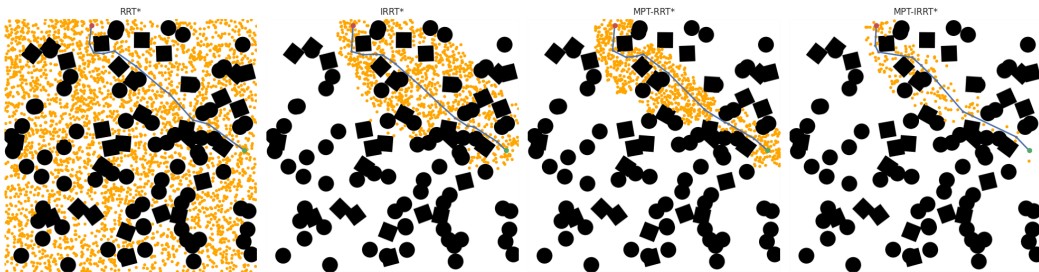

Figure 11: (From left) Path planned by RRT*, IRRT*, MPT-RRT*, and MPT-IRRT* for the same start (green) and goal (red) positions for the Random Forest environment. MPT aided planners are able to reduce significantly the number of vertices (orange) required to search for a path.

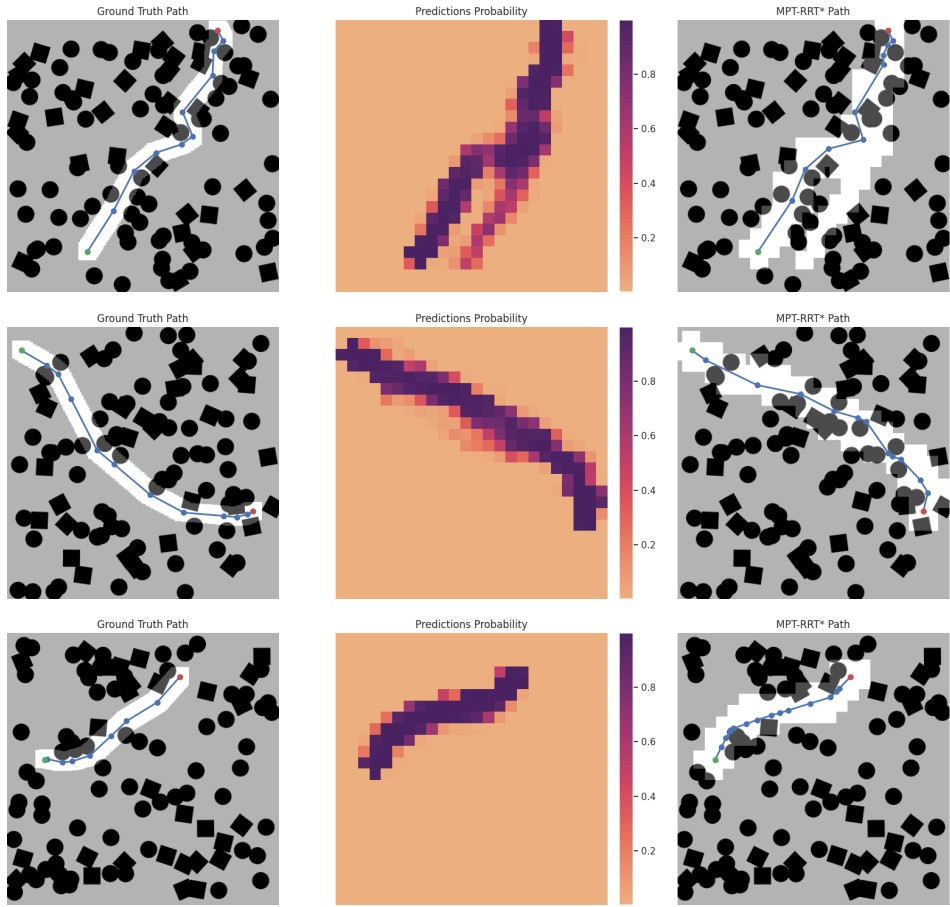

Figure 12: Three different trajectories planned successfully using MPT on the Random Forest environment. Left Column: The ground truth path in the validation dataset. Middle Column: The probability estimate made by the MPT. Right Column: The planned path using RRT* on the region proposed by the MPT.

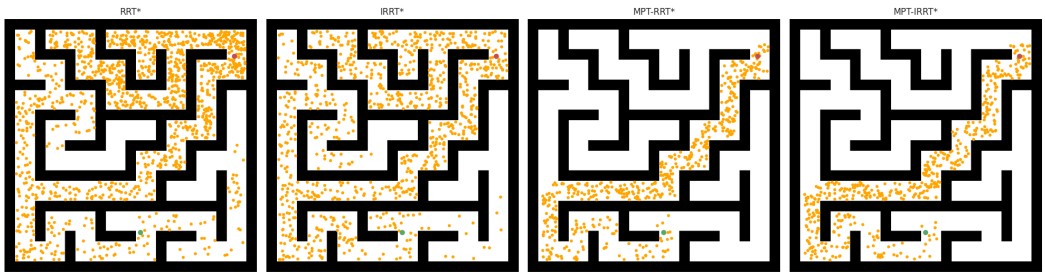

Figure 13: (From left) Path planned by RRT*, IRRT*, MPT-RRT*, and MPT-IRRT* for the same start (green) and goal (red) positions for the Maze environment. MPT aided planners are able to reduce significantly the number of vertices (orange) required to search for a path. The IRRT* planner has to search the entire space for these kinds of maps.

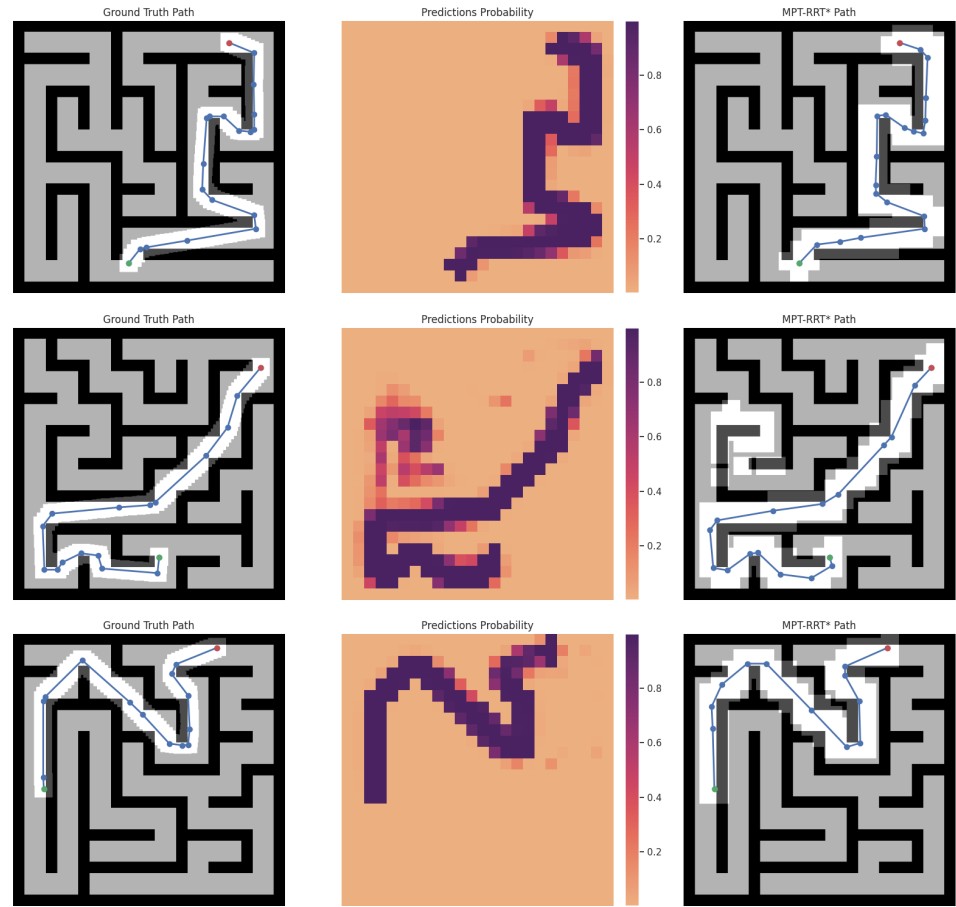

Figure 14: Three different trajectories planned successfully using MPT on the Maze environment. Left Column: The ground truth path in the validation dataset. Middle Column: The probability estimate made by the MPT. Right Column: The planned path using RRT* on the region proposed by the MPT.

## F    DISTRIBUTION OF METRICS FOR MAPS OF DIFFERENT SIZES

The distribution of metrics reported in Table 2 is given in Fig. 15.

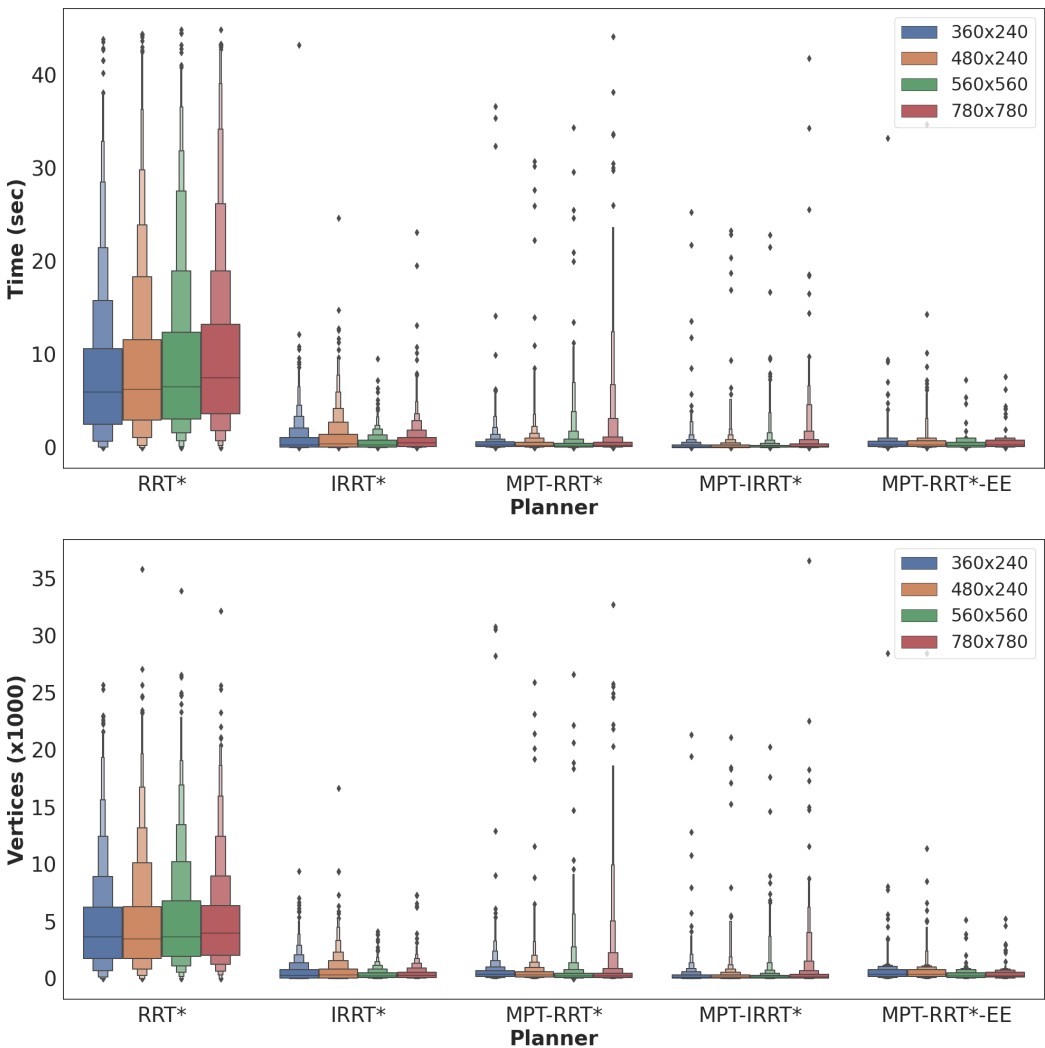

Figure 15: The distribution of metrics for maps of different sizes. Top: The distribution of planning time for different algorithms for maps of different sizes. Bottom: The distribution of the number of vertices in the planning tree for different algorithms for maps of different sizes. MPT aided planners can achieve faster planning times and lower variance in the metrics across maps of different sizes, which were not part of the training data.

# G  FAILED TRAJECTORIES

In Fig. 16 and 17, we visualize some of the trajectories that MPT could not solve for the Random Forest and Maze environment, respectively. These failures were mainly caused by a single patch that was not classified as part of the path, resulting in a map with no valid paths.

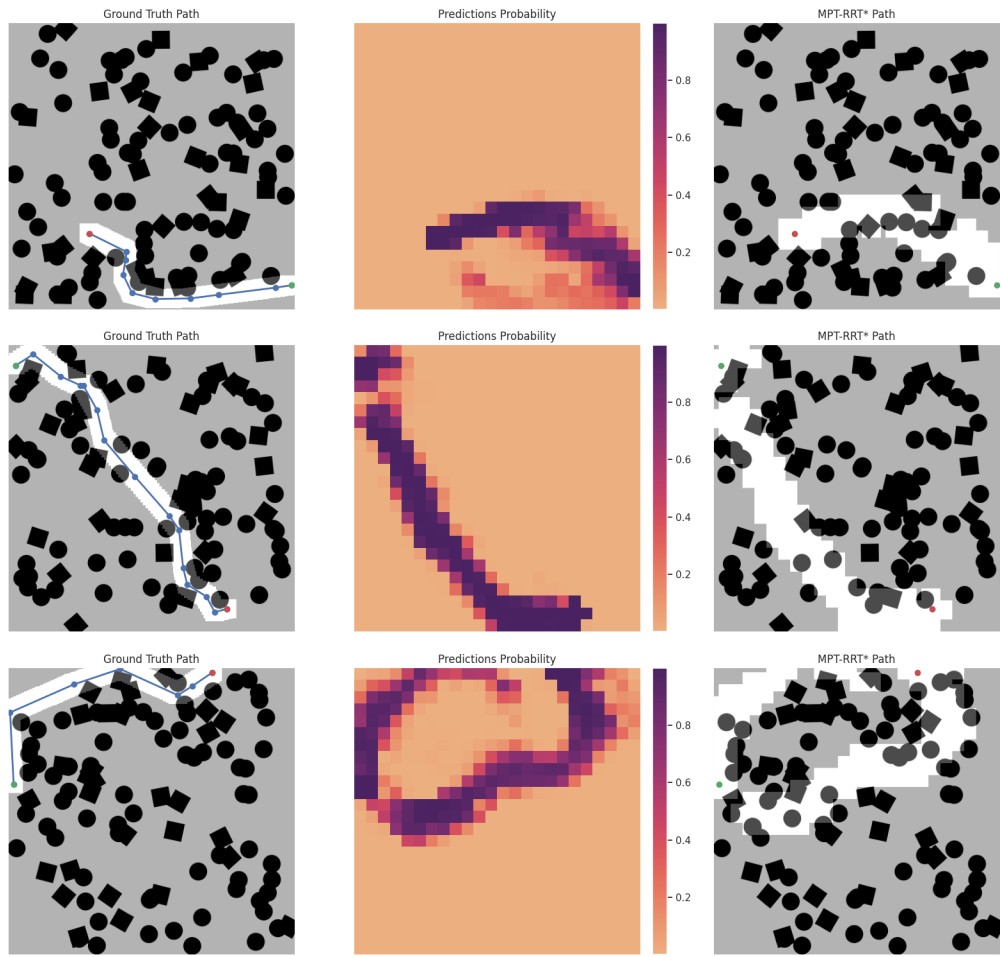

Figure 16: Three different trajectories planned unsuccessfully using MPT on the Random Forest environment. Left Column: The ground truth path in the validation dataset. Middle Column: The probability estimate made by the MPT. Right Column: The planned path using RRT* on the region proposed by the MPT. We observe that in all three cases the failure to predict a few patches resulted in a scenario with no valid paths.

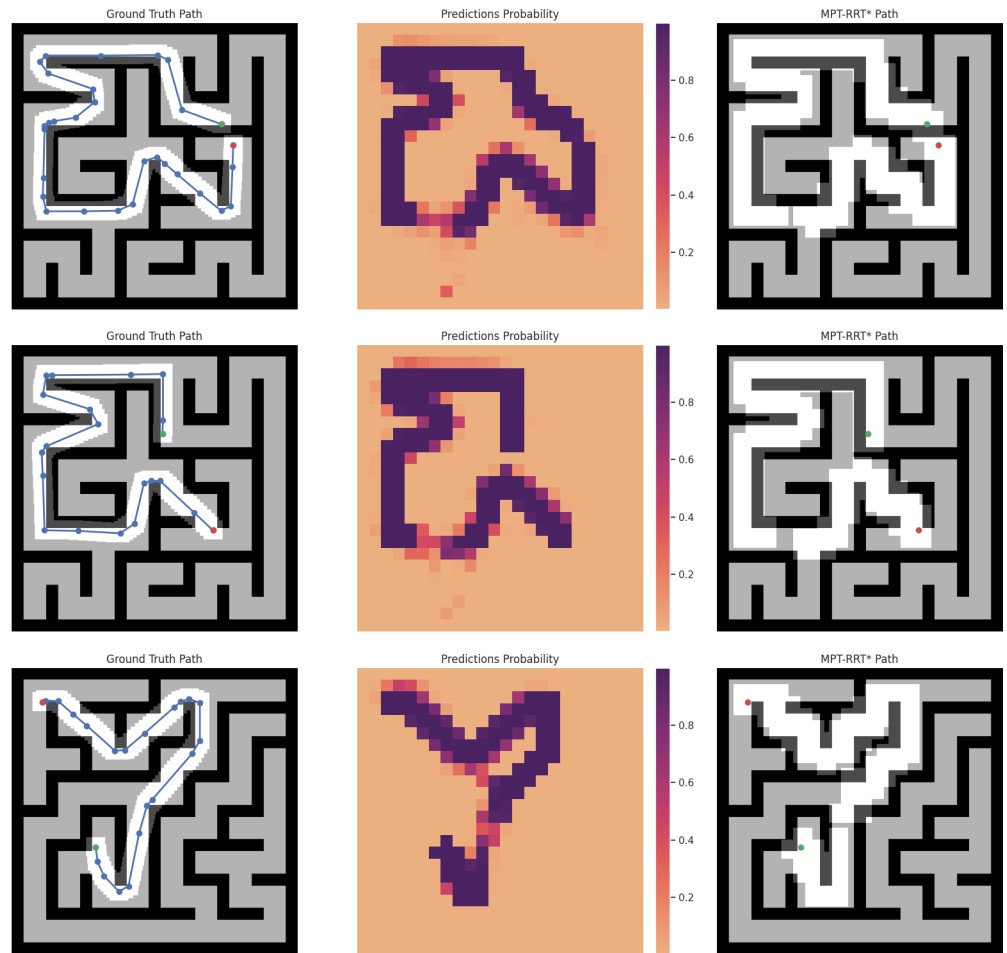

Figure 17: Three different trajectories planned unsuccessfully using MPT on the Maze environment. Left Column: The ground truth path in the validation dataset. Middle Column: The probability estimate made by the MPT. Right Column: The planned path using RRT* on the region proposed by the MPT. We observe that in all three cases the failure to predict a few patches resulted in a scenario with no valid paths.

