# OpenReview forum: "Motion Planning Transformers: One Model to Plan them All"
_ICLR.cc/2022/Conference — ICLR 2022 Submitted_

### Official Review · Reviewer_WoFc · 2021-10-30

**Correctness:** 3
**Technical Novelty And Significance:** 2
**Empirical Novelty And Significance:** 2
**Recommendation:** 6
**Confidence:** 3

**Main Review:**

There are two important aspects of this method. First is to get a region so that a sampling based method can derive the path within that region to save time in the path planning. Second due to the transformer this method can scale to different resolution of map.
The neural guided RRT is not new.  Like "Neural RRT*: Learning-Based Optimal Path Planning" also aim to get a guidance. Similarly "Path Planning using Neural A* Search" which uses different planner but similar philosophy. It would be better to position the advantage of the current method for a fix resolution against these similar existing method which share same philosophy.
The resolution invariant property is nice. It will be good to see the generalisation of this method as it uses a classifier to train. In the paper it is mentioned that the training used both maze and random forest and test on both. To understand what the network learns can we see the result trained on maze and test on random forest or train on both and test on different map with different densities? If the maze has higher density  (means very narrow path/free space) does the method without further training work? Any classification inaccuracy may lead to a completely different and longer path which the author also shows in appendix. So a study on the generalisation ability out of distribution would be good given the IRRT* performance.
In the Table 1, the time exclude the MPT planner which I think is not correct because MPT planner is the key to reduce the RRT* time. The time should be the total time of MPT based mask generation + RRT* planning. Also the path length should also be compared across method. A comparison of time and path length with "Learning to Plan in High Dimensions via Neural Exploration-Exploitation Trees" will be better in a single resolution.
The authors mention about a few choices of the position encoding of the transformer. This is an important choice to take care and a detail ablation of this with the resulted path can be useful to the reader to understand what is happening there.

**Summary Of The Paper:**

This paper proposes a transformer based method for 2D motion planning. The aim of the method is to work in any resolutions of the 2D map and the attention of the transformer enable to get the best possible regions through which the desired path should exist. A  traditional planners is then used given the extracted region to generate the final collision-free path. The results shows the efficacy of the transformer guided path planning using RRT* in different resolution of maps.

**Summary Of The Review:**

The method is good as a pipeline but it needs few careful experiments to position the claims with respect to state-of-the-arts. While part of the method's philosophy is already existed like neural guided path planner, the other part where the resolution invariant is being claimed is good. Some of the comparisons are important to position the entire pipeline against SoA  and few metrics needs to be shown (like the time) to understand the computational complexity.

---

> ### Author Response · Authors · 2021-11-16
> **Discussion and comments**
>
> We thank the reviewer for their thoughtful comments and suggestions. We have addressed some of these issues, such as adding preprocessing time, testing on Neural Exploration-Exploitation Trees to our paper, and have added additional information about the position encoder in the Appendix. Detailed comments are as follows:
>
> >A comparison of time and path length with “Learning to Plan in High Dimensions via Neural Exploration-Exploitation Trees” will be better in a single resolution
>
> We appreciate the reviewer for suggesting papers [1], [2], and [3]. As the reviewer has pointed out, none of these methods evaluate their methods on maps of different dimensions, and even the ones that do, train a separate model for each problem set. Another critical difference between these methods and MPT is the size of the maps used for planning. The smallest map we used for testing is of size 360x240, whereas the largest map size evaluated for all three papers is 128x128. We emphasize larger maps because they capture larger environment spaces and represent finer details such as narrow corridors.
> To bolster the ability of MPT in planning long horizon problems compared to recently proposed planning methods, we modify the Maze environment maps to fit the network proposed in Neural Exploration-Exploitation Trees (NEXT) [3]. We have added the results in Appendix D of our paper. The planner only has an accuracy of 29% and nearly takes 3x the time to plan for a path. We attribute the poor performance of the planner to getting stuck in local minima while planning, whereas MPT-based planners, even with the higher resolution, are able to solve all problems with a fraction of the time.
>
> >To understand what the network learns can we see the result trained on maze and test on random forest or train on both and test on different map with different densities?
>
> We show three kinds of generalizability through our experiments: 1. Generalizability to unseen maps of similar size as the training data. 2. Generalizability to maps of different sizes. 3. Generalizability to real-world maps. The experiment with maps of different sizes and real-world experiments could be seen as out-of-distribution samples since the generated maps did not belong to the distribution as the training data. Our opinion is that the generalizability of MPT to maps of different sizes and real-world maps is more valuable to the community than training the model on a single environment type and testing on maps of different kinds.
>
> >In the Table 1, the time exclude the MPT planner which I think is not correct because MPT planner is the key to reduce the RRT* time. The time should be the total time of MPT based mask generation + RRT* planning.
>
> We agree with the reviewer that the planning time should be included. The MPT preprocessing step adds about 2ms-40ms of overhead time depending on the size of the map and the number of anchor points. We have recalculated our times for all our experiments and updated the timing results up to the third decimal point. Even with the preprocessing time, our results indicated that MPT augmented planner outperforms IRRT*, SST, and other learning-based planners.
>
> >Also the path length should also be compared across method.
>
> For all our experiments, we have a stopping condition for the optimal planner. The planner stops whenever the length of the planned path is equal to or shorter than a trajectory planned by an RRT* planner for 2mins. Because of this, all planners achieve similar results in terms of path length and are nearly optimal.
>
> We hope these experiments and paper modifications have clarified the reviewer's queries. Please let us know if there are any further questions.
>
> [1] J. Wang, W. Chi, C. Li, C. Wang and M. Q. . -H. Meng, “Neural RRT*: Learning-Based Optimal Path Planning,” in IEEE Transactions on Automation Science and Engineering, vol. 17, no. 4, pp. 1748-1758, Oct. 2020, doi: 10.1109/TASE.2020.2976560.
>
> [2] Yonetani, R., Taniai, T., Barekatain, M., Nishimura, M., & Kanezaki, A. (2021, July). Path planning using neural a* search. In International Conference on Machine Learning (pp. 12029-12039). PMLR.
>
> [3] Chen, B., Dai, B., Lin, Q., Ye, G., Liu, H., & Song, L. (2019). Learning to plan in high dimensions via neural exploration-exploitation trees. arXiv preprint arXiv:1903.00070.

---

> > ### Comment · Reviewer_WoFc · 2021-11-19
> > **Clarity of the method and experimentations after the author response leading to the improved recommendation**
> >
> > Thanks authors for clarifying doubts in the experiments. It would have been better to position this paper from the perspective of the path planning efficacy against Neural RRT* for the completeness. The rest of the answers are satisfying and demonstrated the advantages of the method against SoA.

---

### Official Review · Reviewer_J9BR · 2021-11-02

**Correctness:** 4
**Technical Novelty And Significance:** 3
**Empirical Novelty And Significance:** 3
**Recommendation:** 6
**Confidence:** 4

**Main Review:**

The paper is well written, clearly presented and well-motivated. The proposed approach combines learning-based and sampling-based planning approaches using the strength of each approach to solve the task it is best suited to. The authors present several important contributions, from using transformers to ensure the planner can attend the entire map in deciding which areas to sample from, to using positional encoding in a way that allows them to generalise to unseen maps. Furthermore, the approach is presented in a way that makes it reasonable to assume it will generalise to significantly larger maps.

There are a couple of concerns. Firstly, the evaluation presents 2 planning scenarios: point-robot and Dubins car. The results on the point-robot are well presented and thorough, however I was less convinced by the Dubins-path results. It seems like what the MPT has to learn is very similar in both cases, and there is very little evidence that the MPT network is learning any kinematic-specific strategies. It would be interesting to explore this area further, perhaps with maps that have both kinetically feasible and unfeasible trajectories. Ideally, a point-robot MPT would choose trajectories regardless of kinematic feasibility, whereas the Dubins car model would choose the kinematically feasible one. Secondly, the approach seems to be limited to 2D. While I understand that this is a large domain in-and-of itself, it is important to notice that RRT* and its variants are designed for higher-dimensional spaces and can be outperformed by simple A* in 2D grids. It would be interesting to see if the approach can generalise to higher dimensions, making it feasible for robotic arms and more complex planning spaces. It seems like the windowed approach should be extendable to higher dimensions? Finally, the authors do not compare against learning-based planners because they did not perform well, however it would be interesting to see the results for context.

A few minor things:
-Page 4: “For maps larger than then the” -> remove then
-Page 5: “For Dubins Car Model” -> “For the Dubins Car Model”
-Figure 5, consider re-moving RRT* to make the other approaches more easy to read. RRT* is not constrained in sampling, so it makes sense.
-Page 6: Figure 4 is referenced before 3, might be worth re-ordering figures. and 4 are referenced in

**Summary Of The Paper:**

The paper presents a Transformer-based planning approach that attempts to use an attention mechanism to reduce the search-space of a traditional planner such as RRT*. Approaches such as Informed RRT* constrain the search space of the planner, which allows much faster convergence time. In a similar way, the authors present an approach that uses attention on a 2D map to create a "mask" for a standard planner to draw samples from. The proposed Motion Planning Transformer (MPT) works by using sliding window on the original map, creating features that are then positionally encoded (which helps with generalisation) and then put through a transformer encoder and classified. The result is a prediction, for each patch, as to whether a planner should draw samples from it. When combined with a planner it is able to achieve state-of-the-art results in a reduced time.

**Summary Of The Review:**

The paper presents a solid approach to pathplanning that solves the problem in an interesting way. There is a good amount of novelty and  experimental evidence and the approach is technically sound.

---

> ### Author Response · Authors · 2021-11-16
> **Discussions and comments**
>
> We thank the reviewer for their thoughtful comments and suggestions, particularly for taking the time to point out the minor mistakes. We have addressed these errors and added a new comparison to the Neural Exploration-Exploitation Trees [1] for comparing against a value-driven search method. A brief discussion on other queries is detailed below:
>
> >Showing proof that the MPT learns kinematic-specific strategies
>
> The MPT-based model does incorporate the kinematics into the highlighted region. Take, for example, the planning problem shown on the left in Fig 8. For point robots, the optimal path is much different from the one taken by the Dubins car model. An optimum plan would be to steer through the dense obstacle space rather than go around it. Since such a plan would not be feasible for the kinematically constrained robot, MPT highlights a region that circumvents the obstacles.
>
> >Why not higher dimensional robots?
>
> When extending this method to a higher dimensional robot, new sets of challenges arise. The first being, the representation of the planning scene using points clouds or voxels, which are intractable on the current Transformer models due to memory constraints. Another hurdle to cross would be to learn the mapping between environment and planning space. Since most higher-dimensional robots are manipulators, the current formulation needs additional machinery to adapt to such planning problems. We believe this is outside the scope of this work and is an interesting future direction of the current method.
>
> We hope our discussion has clarified the reviewer’s queries. Please let us know if there are any further questions.
>
> [1] Chen, B., Dai, B., Lin, Q., Ye, G., Liu, H., & Song, L. (2019). Learning to plan in high dimensions via neural exploration-exploitation trees. arXiv preprint arXiv:1903.00070.

---

> > ### Comment · Reviewer_J9BR · 2021-11-22
> > **Kinematics, Timings and Higher Dimensions**
> >
> > Thank you to the authors for clarifying my questions. I believe the paper is good and presents an interesting way forward, but there are still important questions to be answered, specifically regarding kinematics (as the method has no mechanism to optimise kinematics). There is also a concern about the additional machinery required to expand this to higher dimensions, such as constraining the orientation of the robot. I have updated my review to reflect this.

---

### Official Review · Reviewer_JNUy · 2021-11-03

**Correctness:** 3
**Technical Novelty And Significance:** 4
**Empirical Novelty And Significance:** 4
**Recommendation:** 6
**Confidence:** 4

**Main Review:**

Strengths:
+ Reduced planning time over traditional approaches (there are some issues here that I talk about in the weaknesses).
+ Improved performance over learning based approaches.
+ Method does not need fixed input size as other learning based approaches.
+ Well written paper.

Weaknesses / Comments / Questions:

- Time to pass through MPT:
The key weakness in this work is not counting the time it takes to MPT as the total processing time. The authors argue that this can be seen as a preprocessing step, however, I think this may not be a valid argument. The entire method should be timed in order to compare to the other baselines to have a fair comparison. I would like to see these times to conclude whether this method in fact pushes the envelope against traditional methods. If the baselines have pre-processing steps, these times should be included as well for fairness.

- Using the hybrid approach with learning based baselines:
This author proposes the use of a hybrid approach to addressed any trajectories that miss reaching the goal. The authors use this with their proposed method, however, it is not used with the other learning based approaches. I think this experiment is necessary for completeness to see whether the MPT is a key component in the performance and it could not just be replaced, say, with the MPnet or UNet baselines.


- Sampling k during training:
I don't quite get the reasoning behind sampling k for the positional encodings other than to make the training noisy, and thus, difficult for the model to memorize the positions in the grid. If this is the case, I would encourage the authors to investigate this further and maybe find a more mathematical explanation for this type of regularizer which obviously seems to work.


- Method marginally outperforms baselines in the "Dubins car model" experiments (Table 4).
This goes back to measuring the time it takes to run MPT. What happens if this time is included? Is the baseline faster than the proposed method?

**Summary Of The Paper:**

This paper proposes a transformer based trajectory estimator for 2D navigation. The proposed method is shown to not be limited by input size as other learning based approaches. In the experimental section, the authors show that their estimated trajectories result in speed up and performance boost for planning in comparison to traditional and other learning approaches.

**Summary Of The Review:**

I personally like this approach, but I feel that there are key missing questions that the paper does not answer in order to determine whether the claims are successfully substantiate or not. I am willing to increase my score depending on whether the author's response address my questions.

---

> ### Author Response · Authors · 2021-11-16
> **Discussions and comments**
>
> We thank the reviewer for their thoughtful comments and suggestions. We have addressed some of the key concerns, including preprocessing time into the planning time, testing hybrid methods for other learning-based methods, and additional information for k-means sampling into the paper.
>
> >Time to pass through MPT, Dubins car results
>
> The MPT preprocessing step adds about 2ms-40ms of overhead time depending on the size of the map and the number of anchor points. We have recalculated our times for all our experiments and updated the timing results up to the third decimal point. Even with the preprocessing time, our results indicated that MPT augmented planner outperforms IRRT*, SST, and other learning-based planners.
>
> >Using the hybrid approach with learning based baselines
>
> We have added a hybrid version of UNet-RRT*. UNet-RRT*-EE achieves 100% accuracy, but the timing and vertex count is similar to that of RRT*, which indicates that the plan for the path relies more on RRT* rather than the region marked by the UNet model. The MPNet method, on the other hand, already relies on RRT* for proposing an alternative plan when the model fails; hence it requires no additional exploration. Thus the hybrid approach allows for higher accuracy of the MPT-RRT*-EE method while maintaining similar times and vertex counts as MPT-RRT*. We also compared the Neural Exploration-Exploitation Tree (NEXT) [1] method by modifying our maze environment. Since we compare the method with a simplified map, we report the results in the Appendix (Section D).
>
> >Sampling k during training
>
> We observed that the MPT model trained on the position encoding proposed by Vaswani et al. (2017) showed signs of overfitting. The model fails to highlight the region outside the training domain (See Figure 9(Left ) in Appendix for an example). One way to resolve this overfitting is to train on maps of various sizes to ensure that the network generalizes to maps of different position encoding. However, this procedure is often computationally expensive, especially for larger maps. We propose randomly translating the planning space so that the network observes different sets of position encoding (see Figure 9 (Right)). $k$ is randomly chosen to have the effect of a translated map. By training this way, we avoid the model from overfitting to the training data and generalize to maps of different sizes.
>
> We hope these experiments and paper modifications have clarified the queries. Please let us know if there are any further questions.
>
> [1] Chen, B., Dai, B., Lin, Q., Ye, G., Liu, H., & Song, L. (2019). Learning to plan in high dimensions via neural exploration-exploitation trees. arXiv preprint arXiv:1903.00070.

---

> > ### Comment · Reviewer_JNUy · 2021-11-29
> > **Rebuttal**
> >
> > I would like to thank the authors for their response. The authors addressed most of my concerns, and so, I am increasing my original score. I am still not fully convinced by the small time gap between the baseline and the proposed method in Table4, but maybe this could be addressed in the future.

---

### Official Review · Reviewer_jDfv · 2021-11-04

**Correctness:** 3
**Technical Novelty And Significance:** 2
**Empirical Novelty And Significance:** 2
**Recommendation:** 3
**Confidence:** 4

**Main Review:**

The paper tackles an important problem of learning to plan which is of interest to the machine learning community and has important applications in robotics. The writing quality is good and the paper is easy to follow.

The authors conduct experiments with large map sizes and show effectiveness of the proposed approach. The performance of the method drops significantly when map sizes are increased, which indicates that the method is not scalable. This weakness in itself is not very significant as prior work has experimented with much smaller map sizes.

The bigger weakness is that the proposed method is not differentiable, which is one of the main benefits of prior learning-based planning methods such as VINs over traditional planners. Differentiability allows the planning models as a component in larger end-to-end models (for example in CMP (Gupta et al., CVPR 2017)). It seems that the only benefit of the proposed approach over traditional planners is that it is faster. However, the lack of differentiability combined with poor scalability means that the proposed approach is only useful for real-time planning on small maps, and even in this case, the proposed approach performs slightly worse than traditional planners (100% vs 99.2%). This makes the significance of the approach very low in my opinion. Traditional planners would be still be preferred over the proposed method when maps are large or when performance is more important than runtime.

The use of Transformers for planning is not well motivated in the submission. Prior work has utilized convolutional neural networks and convolutional LSTMS for this problem, but the authors have not provided empirical comparisons to those methods. If the motivation is to tackle variable map sizes, why not use recurrent neural networks? Comparison with recurrent and convolutional neural networks is important to establish the significance of Transformers for this problem. The authors have added a convolutional network-based baselines (UNet) in certain experiments but there are no details available on how this baseline is implemented.

The authors only conduct experiments for 2D navigation tasks. It is unclear whether the approach would scale to manipulation tasks with higher degrees-of-freedom.

**Summary Of The Paper:**

This paper proposes a learning-based method to solve planning problems. The approach first identifies regions on the map using transformers to provide attention to map areas likely to include the best path, and then applies local planners to generate the final collision-free path. Experiments show that the proposed method can achieve performance comparable to the traditional planners with lower planning time for smaller maps, but the performance drops significantly as the map sizes increase.

Note that I previously reviewed this paper for NeurIPS 2021. The authors have made a few changes since the NeurIPS version and I have updated my review accordingly.

**Summary Of The Review:**

Although the paper tackles an important problem, it has several weaknesses in terms of motivation and missing baselines. Lack of differentiability and poor scalability severely limits the applications of the proposed approach.

---

> ### Author Response · Authors · 2021-11-16
> **Discussions and comments**
>
> We want to thank the reviewer for once again taking their time to review our paper. In this current submission, we have improved upon the previous submission in the following ways:
> 1. Added random position encoding to improve results for maps of larger sizes.
> 2. Compared our method with other learning-based planners.
> 3. Restructured our paper to motivate the use of Transformers.
>
> In the following, we reply to some of the other comments. It would be greatly appreciated if the reviewers would actively participate in this discussion since we did not receive a reply in our last submission.
>
> >The performance of the method drops significantly when map sizes are increased, which indicates that the method is not scalable.
>
> MPT-based planners achieve nearly 99% accuracy, while the hybrid planner achieves 100% accuracy for larger maps. We believe that it is not justified to call the drop in a single percentage “significant.”
>
> >The bigger weakness is that the proposed method is not differentiable
>
> It is unclear what the reviewer means by “differentiable” here. If they are referring to the ability of Value Iteration Networks (VINs) [1] to embed the value iteration within the network architecture, then it is not a necessary condition to implement planning models within larger end-to-end models. An example of this would be [2]. The current formulation of the model does not prohibit using the method as a component in larger end-to-end models, such as a reactive policy. Since the map generated by the MPT represents regions/states on which the reactive policy should focus, i.e., attend to. Although this study is interesting and worth looking into, we believe that it is outside the scope of this paper.
>
> >It seems that the only benefit of the proposed approach over traditional planners is that it is faster
>
> Through our experiments, we show that MPT augmented planners:
> 1. Achieve 7-28% improvement in accuracy over recent learning-based planners
> 2. Achieve similar accuracy to traditional planners.
> 3. Reduce planning time by 7-25x and the vertices on the planning tree by 2-12x compared to traditional planners.
> 4. Generalize to maps of different sizes while maintaining accuracy and reducing planning time and vertices in the planning tree.
> 5. Generalize to o real-world environments without fine-tuning or additional training.
>
> Hence MPT has numerous other benefits over traditional and learning-based planners.
>
> >The use of Transformers for planning is not well motivated in the submission.
>
> We believe that the second paragraph in the introduction section addresses this specific query. For brevity, we reiterate the same here.
> >>The trajectory of a local plan is influenced by the orientation of far-away obstacles, similar to language models where the semantics of a sentence is inferred only by reading the entire sentence. The ability of transformers to understand the syntactic and semantic structure of sentences was shown [3]. Thus, the capacity of transformers to learn such connections efficiently is the motivation of this work.
>
> >The authors have added a convolutional network-based baselines (UNet) in certain experiments but there are no details available on how this baseline is implemented.
>
> The UNet model is the same segmentation network proposed in [4]. The input to the UNet is the same as the one used for MPT. The network architecture is provided in the Appendix section.
>
> >The authors only conduct experiments for 2D navigation tasks. It is unclear whether the approach would scale to manipulation tasks with higher degrees-of-freedom.
>
> We only claim that the proposed model is used for 2D navigation and make it clear from the very start. Although the extension of the current method towards manipulation tasks is an interesting problem, as mentioned in the concluding statements, it requires further study and is not in the purview of this paper.
>
> We hope this discussion has clarified the reviewers’ queries. Please let us know if there are any further questions.
>
> [1] Tamar, A., Wu, Y., Thomas, G., Levine, S., & Abbeel, P. (2016). Value iteration networks. arXiv preprint arXiv:1602.02867.
>
> [2] Banino, Andrea, et al. “Vector-based navigation using grid-like representations in artificial agents.” Nature 557.7705 (2018): 429-433
>
> [3] Vaswani, A., Shazeer, N., Parmar, N., Uszkoreit, J., Jones, L., Gomez, A. N.,. & Polosukhin, I. (2017). Attention is all you need. In Advances in neural information processing systems (pp. 5998-6008).
>
> [4] Ronneberger, O., Fischer, P., & Brox, T. (2015, October). U-net: Convolutional networks for biomedical image segmentation. In International Conference on Medical image computing and computer-assisted intervention (pp. 234-241). Springer, Cham.

---

### Decision · Program_Chairs · 2022-01-20

**Decision:**

Reject

**Comment:**

The paper proposes a planning framework that uses a transformer-based architecture as an attention mechanism that guides the search of a traditional sample-based planner (e.g., RRT*). More specifically, features extracted from a sliding window over the 2D search space serve as input to a transformer that produces a mask indicating where to draw samples from. By constraining the search space for the sample-based planner, the method reduces the time required for planning. The method is compared to both traditional and learning-based planners on different 2D navigation tasks and found to improve sample complexity (and, in turn, computation time), while also being capable of generalizing to unseen and real-world maps.

The manner by which the method combines the advantages of sample-based planning with an attentional mechanism as a way to constrain the sampling process is interesting. As the reviewers emphasize, the experimental evaluation shows that this approach results in performance gains over both traditional (sample-based) and learning-based planners, while also being able to scale to larger maps as well as better generalize to out-of-distribution settings (compared to learning-based methods). These results support the value of both the overall approach as well as the architectural components (e.g., the transformer and the use of positional encoding). The reviewers initially raised a few concerns with the paper, the most notable of which are the need to include preprocessing in the overall computation time, the accuracy of some of the claims in the paper (e.g., with regards to generalizability), generalization to higher-dimensional domains, and the performance on the Dubins car domain. The authors responded to each of the reviews and updated the submission to address many of these concerns. However, questions still remain regarding whether or not the approach can be adapted to state/configuration spaces with more than two dimensions, something that traditional planners are readily capable of, and the unconvincing results on the Dubins car domain.

Overall, the paper proposes an interesting approach to an important problem that is relevant to the robotics and machine learning communities. The paper makes promising contributions to improve the efficiency of planning, however the significance of these contributions needs to be made clearer.